# The Global Impact of Hepatitis B Vaccination on Hepatocellular Carcinoma

**DOI:** 10.3390/vaccines10050793

**Published:** 2022-05-17

**Authors:** Joan Ericka Flores, Alexander J. Thompson, Marno Ryan, Jessica Howell

**Affiliations:** 1Department of Gastroenterology, St. Vincent’s Hospital Melbourne, Fitzroy, VIC 3065, Australia; alexander.thompson@svha.org.au (A.J.T.); marno.ryan@svha.org.au (M.R.); jessica.howell@svha.org.au (J.H.); 2Department of Medicine, University of Melbourne, Parkville, VIC 3010, Australia; 3Disease Elimination Program, Burnet Institute, Melbourne, VIC 3004, Australia; 4Department of Epidemiology and Preventative Medicine, Monash University, Clayton, VIC 3800, Australia

**Keywords:** hepatitis B, vaccination, hepatocellular carcinoma, public health

## Abstract

Over 1.5 million preventable new hepatitis B infections continue to occur each year and there are an estimated 296 million people living with chronic hepatitis B infection worldwide, resulting in more than 820,000 deaths annually due to liver cirrhosis and hepatocellular carcinoma (HCC). Hepatitis B vaccination remains the cornerstone of public health policy to prevent HCC and a vital component of the global hepatitis B elimination response. The WHO has set a 90% vaccination target to achieve hepatitis B elimination by 2030; however, there is wide variability in reported birth dose coverage, with global coverage at only 42%. In this review, we outline the global trends in hepatitis B vaccination coverage and the impact of hepatitis B vaccination on HCC incidence and discuss the challenges and enabling factors for achieving WHO 2030 hepatitis B vaccination coverage targets.

## 1. Introduction

Hepatitis B and its sequelae of liver cirrhosis and hepatocellular carcinoma (HCC) are preventable by vaccination, yet the global burden of hepatitis-B-related morbidity and mortality remains unacceptably high. As of 2019, over 1.5 million preventable new infections continue to occur each year and there are an estimated 296 million people living with chronic hepatitis B infection, resulting in more than 820,000 deaths annually due to liver cirrhosis and hepatocellular carcinoma (HCC) [1]. HCC is one of the fastest increasing causes of cancer death and hepatitis B is the cause of a third of all liver cancer deaths worldwide [2,3,4]. Importantly, hepatitis-B-related HCC disproportionately affects low resource countries, accounting for two thirds of primary liver cancer cases, compared with a quarter of HCC cases in more developed countries. In light of the availability of direct acting antiviral (DAA) therapies and the global reduction in hepatitis-C-related HCC incidence and mortality [5,6], the proportion of HCC due to hepatitis B will increase.

Hepatitis B is a class I carcinogen and chronic infection carries a 10–25% lifetime risk of developing HCC, directly through integration of hepatitis B covalently closed DNA (cccDNA) into the host hepatocyte genome and activation of genes that promote cancer development; and indirectly through immune activation, leading to inflammatory hepatocellular injury, fibrogenesis and hepatocyte regeneration [7]. HCC risk is associated with persistent infection and there is typically a lag time of thirty to forty years between infection onset and HCC development [7,8,9].

Children infected by hepatitis B are more likely to develop chronic infection, with risk of chronicity determined by age at infection. The vast majority of children infected within the first 12 months of life (80–90%) develop chronic hepatitis B, compared with 20–30% when infected during early childhood and 5% in adulthood [10,11]. The timing of hepatitis B transmission also determines the risk of HCC development: mother to child (vertical) transmission is associated with an increased risk of HCC compared with transmission between children and family members early in childhood (horizontal transmission) [12]. Compared with first-order children born to mothers with higher levels of viremia, the odds of developing HCC were between 48 and 86% lower in subsequent-order children born to the same HBsAg positive mothers with lower levels of viremia [13]. In this way, early vaccination against hepatitis B from the time of birth is vital for reducing transmission and subsequent persisting infection that results in oncogenesis.

First developed in the 1970s, the HBV vaccine is highly effective at preventing vertical hepatitis B transmission when the full three or four dose vaccination schedule is given at birth and in early infancy; this in turn prevents onward transmission, liver cirrhosis and HCC [10]. The hepatitis B vaccine is therefore the first cancer-preventing vaccine [12,14]. Hepatitis B vaccination remains the cornerstone of public health policy to prevent HCC and a vital component of the global hepatitis B elimination response. The WHO has set a 90% vaccination target to achieve hepatitis B elimination by 2030.

In this review, we outline the global trends in hepatitis B vaccination coverage and the current and projected impact of hepatitis B vaccination on HCC incidence. We present real-world data on HCC incidence reduction in high endemicity areas that were early adopters of universal hepatitis B vaccination programs. We also discuss the challenges and enabling factors for vaccination coverage to achieve the WHO 2030 hepatitis B elimination targets.

## 2. The Impact of Hepatitis B Vaccination on HCC Incidence: Real-World Data

Globally, there has been a decrease in hepatitis-B-related primary liver cancers, attributed to hepatitis B vaccination [15]. Although the impact of universal vaccination programs on HCC prevention may take some time to quantify due to the long latency of HCC development, several longitudinal studies from Asia, The Gambia and within specific hepatitis B endemic populations have already demonstrated the positive impact of universal hepatitis B vaccination on hepatitis-B-related childhood HCC incidence (Table 1, Appendix A).

### 2.1. Taiwan

Within Taiwan, legislation for vaccination was developed in 1984. From 1984 to 1986, hepatitis B vaccination was provided to infants born to hepatitis B surface Ag positive mothers. After 1986, universal hepatitis B vaccination was scaled up to all infants, initially with the plasma-based vaccine from 1986 to 1992, then the recombinant Hepatitis B vaccine after July 1992 [16]. Early analysis of primary liver cancer incidence and mortality data from the Taiwan National Cancer registry and National Mortality registry from 1981 to 1994 conducted by Chang et al. [16] noted a significant decrease in the incidence of HCC in children aged 6 to 14 years, with an age-adjusted RR of 0.33 for HCC development and RR of HCC-related death of 0.51 after 1990 (*p* < 0.001) [16].

Assessment of Taiwan’s HCC-related mortality data between 1974 and 1999, as a surrogate for incidence, was conducted by Lee et al. [17]. They observed a significant decrease in mortality in the 0–14 year age group, with 70% and 62% reduction in male and female mortality, respectively, in 1996–1999 compared with 1980–1983 [17]. The incidence rates of HCC development in all age groups born in 1984–1998 following the universal vaccination program had significant decreases in incidence rates (per 100,000 person-years) ranging 0.15–0.19 compared with 0.49–0.60 in the same age groups in 1973–1979 [18]. Follow up of these cohorts into the early 2000s continued to show significant reductions in hepatitis-B-related HCC incidence rates, and the relative risk of HCC development in the vaccinated cohort compared with the unvaccinated cohort ranged from 0.31 to 0.38 across the age groups of 6–9 years, 10–14 years and 15–19 years (*p* < 0.001) (Figure 1) [19].

Follow up of 3,836,988 vaccinees in the year 2000 identified 49 incident HCC cases. Moreover, one of the key risks for HCC was incomplete vaccination, defined as less than three vaccination doses received since the program’s implementation; this had a hazard ratio of 2.52 (95% CI 1.25–5.05; *p* = 0.0094) after adjusting for male sex and maternal HBeAg serostatus [21].

As of 2013, vaccination coverage of infants in Taiwan was very high at 98% [20], and during the period of 2003–2011, there was a significantly decrease in the proportion of HCC due to hepatitis B diagnosed in children and young adults under 29 years to less than 1%, with the majority of hepatitis-B-related HCC cases now occurring in middle-aged adults and elderly people born before the introduction of universal infant hepatitis B vaccination [20]. Notably, over the same time period between 2003–2011 (almost thirty years after the introduction of the universal Hepatitis B vaccination program in Taiwan), there has been a significant annual percentage decrease (*p* < 0.05) in HCC incidence rates from all causes to 16.6%, 7.9% and 2% in the age groups 0–14 years, 15–29 years, and 30–64 years, respectively [20]. More recent follow-up data to 2017 have demonstrated that the cohort aged 30 years and under, who received universal infant hepatitis B vaccination, had a 35.9% reduction in incidence of HCC compared with before the program’s implementation [22].

Collectively, these data suggest that although other factors have likely contributed to reductions in HCC incidence in Taiwan over time, hepatitis B vaccination coverage is a key driver of reduced HCC incidence and mortality.

### 2.2. China

In China, routine immunization of all infants was introduced early in 1992, with the first dose administered within 24 h of birth; however, the roll out of the program was not consistent through the country, with some counties not partaking until the 2000s [23,24]. The mortality from HCC was evaluated in two similar counties [28]: BinYang and LongAn, both rural counties in the Guangxi province. LongAn was host to one of the clinical trials where all newborn infants born between 1986 and 1996 were vaccinated. In contrast, BinYang, a neighboring industrial and affluent county, did not adopt universal vaccination of infants until 2002 and served as an appropriate comparison non-vaccinated group.

HCC-related mortality data from both regions were obtained from the national mortality surveillance system in 2017–2018: LongAn had a significantly higher age-adjusted rate of HCC related mortality of 53.3 per 100,000 py compared with 45.3 per 100,000 py in BinYang between 2017 and 2018 (*p* = 0.005). When childhood mortality within LongAn was assessed before and after the vaccination clinical trial was commenced, HCC mortality in the 20–29 year age group in 2004 among individuals who were not vaccinated at birth had a significantly higher HCC-related mortality death rate (per 100,000) of 7.9 compared with 1.4 in 2017–2018 (*p* = 0.018). This study highlights the decline in HCC associated with the universal immunization of newborns [23].

The Qidong Hepatitis B Intervention Study, a cluster randomized controlled trial of vaccination versus no vaccination (placebo) in 41 rural towns in the Qidong province conducted between 1985–1990 and followed up to 2013, also demonstrated a protective effect of universal hepatitis B vaccination on the development of primary liver cancer, with a hazard ratio of 0.16 in the vaccinated cohort (95% CI 0.01, 0.77, *p* = 0.0224) [24].

### 2.3. The Gambia

Similarly, the Gambian Hepatitis Intervention Study was designed as a randomized controlled step-wedge trial of infant hepatitis B vaccination to determine the impact of their national immunization program in 1986 on chronic hepatitis B prevalence. This study showed a high degree of protection against primary infection and chronic carriage rate [25,29,30,31,32]. The effect of the program on HCC prevention is expected to show statistically significant differences in HCC development from the year 2017 [26,31,33]. The outcomes are yet to be reported.

### 2.4. Alaska

The impact of hepatitis B vaccination on hepatitis-B-related HCC incidence has also been demonstrated in key high-risk populations within low endemicity countries. For example, an Alaskan cohort study in individuals aged less than 20 years in 2010, who received infant vaccination, showed a decline in incidence of childhood HCC to 0 cases per 100,000 population since 1999, from a peak incidence rate of 3 per 100,000 in 1984–1988, indicating the elimination of childhood HCC associated with hepatitis B by vaccination [27].

## 3. The Impact of Hepatitis B Vaccination on HCC Incidence-Modelling Data

The long latency period from infection in childhood to development of hepatitis-B-related HCC later in life means evidence for the impact of vaccination on HCC incidence is only now emerging. In this setting, mathematical modelling is a useful tool to provide estimates of future impact and cost-effectiveness of an intervention such as hepatitis B vaccination on HCC incidence. Early models of the impact of hepatitis B vaccination on HCC incidence in The Gambia Hepatitis Trial estimated a 91% reduction in HCC in a fully vaccinated cohort compared with those unvaccinated, under the assumption that hepatitis B accounted for 90% of attributable risk to HCC development [25]. In Japan, where the vaccination program for newborns with HBsAg-positive and HBeAg-positive mothers was implemented in 1985, modelling projected an observed decrease of 1600 HCC cases in males and 500 HCC cases in females from 2030 to 2050 [34]. Later models utilizing chronic hepatitis B prevalence and HCC incidence data from high endemicity areas (Alaska, China, The Gambia and Taiwan) in the year 2000 estimated that 90% global coverage of the primary series of three vaccination doses would prevent 84% of global hepatitis-B-related deaths from cirrhosis and hepatocellular carcinoma, and 100% vaccination coverage would prevent 95% of hepatitis-B-related deaths [35].

More recently, a global model of hepatitis B elimination conducted by Nayagam et al. [36] estimated that without a change in the status quo of vaccination programs worldwide, 25 million new cases of chronic hepatitis B infection would occur between 2015 and 2030. In contrast, scale-up of infant vaccination global coverage to 90% would avoid 4.3 million incident chronic hepatitis B infections during the same time frame and would prevent 1.1 million hepatitis-B-related deaths by 2030. Additionally, scaling up efforts to improve coverage of primary series vaccination to 90% in combination with birth dose vaccination coverage of 80% is estimated to prevent 1.5 million deaths due to liver cancer by 2030 [36].

## 4. Hepatitis B Vaccination: Current Recommendations and Global Coverage

### 4.1. Hepatitis B Vaccination Preparations and Usage

Hepatitis B vaccines are available as either recombinant monovalent formulations, or pentavalent formulations in combination with diphtheria, tetanus, pertussis, *haemophilus influenzae type b* and inactivated polio vaccines. The pentavalent formulations are most commonly used for the infant vaccination schedule, whereas the monovalent vaccine is used for birth doses or for adults at risk [10]. The current recommended schedule is for all infants to receive the hepatitis B vaccine within the first 24 h of life, regardless of maternal hepatitis B status or community prevalence levels, with two to three subsequent doses each given 4 weeks apart, including booster doses recommended for people who have not received the three-dose series [1].

As of 2020, WHO recommendations have extended to include the commencement of antiviral prophylaxis in the third trimester of pregnancy to reduce rates of mother-to-child transmission, in pregnant women with elevated hepatitis B DNA levels ≥ 200,000 IU/mL (≥5.3 log10 IU/mL) and the administration of hepatitis B immunoglobulin (HBIG) at birth, where available, to infants born to mothers with hepatitis B surface antigen positivity [37]. These recommendations acknowledge issues of availability and cost, and place emphasis on the importance of three-dose hepatitis B vaccination [37].

The WHO Global Health Sector Strategy on Viral Hepatitis has set the target of achieving 90% hepatitis B vaccination coverage, including birth dose, by 2030 [38]. The WHO have recommended universal childhood immunization for hepatitis B since 1991, with the Expanded Program on Immunization (EPI) supporting administration of the ‘primary series’ of three vaccinations to be delivered at 6, 10 and 14 weeks following birth [39]. The addition of the ‘birth dose’ hepatitis B vaccine to be administered within 24 h of birth was later recommended in 2004 to further reduce the risk of perinatal transmission of hepatitis B [40]. In 2012, Gavi supported the distribution of the hepatitis B pentavalent vaccine to assist in improving the coverage of the primary series of infant vaccination [41], with subsequent support for the additional monovalent hepatitis B vaccine from early 2021 [42].

Universal vaccination strategies are considered to be more effective due to the high risk of hepatitis B transmission vertically and also horizontally between children less than five years old, as targeted vaccination would not reduce transmission risk for this group [43]. This is of particular concern in high endemic regions including East Asia, the Pacific Islands and Territories, and sub-Saharan Africa, and intermediate endemicity regions such as Eastern Europe and the Mediterranean [3,44]. Moreover, assessing maternal seropositivity can be an additional challenge to timely birth dose delivery that is overcome by a universal vaccination approach [45]. Transmission in low endemicity regions, such as North America and Western Europe, occurs predominantly via sexual and percutaneous means during early adulthood [46,47] and vaccination is still effective at preventing transmission in adulthood. However, with increasing migration from high to low prevalence areas, adoption of universal hepatitis B vaccination is a logical public health measure [48,49,50].

### 4.2. Global Hepatitis B Vaccination Coverage

There has been a tremendous global effort to achieve high hepatitis B vaccination coverage over the last two decades. Uptake of universal Hepatitis B infant vaccination has been adopted by the 190 WHO Member States as of the end of 2020, with an estimated global coverage of 83% for the three-dose hepatitis B vaccination schedule (Figure 2), and 113 WHO Member States have implemented birth dose vaccination within the first 24 h of life [44,51].

There is wide variability in reported birth dose coverage, from as high as 84% coverage in the Western Pacific Regions to as low as 6% in the African region, and global coverage is only 42% [51] (Figure 3).

In 2015, the South-East Asian Region (SEARO) had an overall increase in three-dose hepatitis B vaccination coverage, up from 56% in 2011 to 87%. As of 2015, the majority of the SEARO countries had achieved the WHO 2030 hepatitis B vaccination target with coverage rates of >90%, with the lowest coverage reported in Myanmar and Timor-Lest (75% and 76%, respectively) [52].

The Western Pacific (WPRO) region has also achieved great success increasing hepatitis B vaccination coverage through early investment in hepatitis B vaccination by several key member states: by 2020, China had an estimated three-dose vaccination coverage of 99%, Mongolia 96% and Vietnam 94% [53]. Early data from Taiwan demonstrated a reduction in prevalence from 9.8% at the time of program implementation in 1984 to 1.3% over its first ten years [54]. Within the Pacific Region Islands and Territories, vaccination coverage in the Cook islands, Fiji, Kiribati, Samoa, Tokelau, Tonga, Tuvalu and Vanuatu has shown major improvements over time with the support of the Control of Hepatitis B Infection in the Pacific Island Countries Project for project supplies and technical support from 1994 to 1998 [55]. Coverage across the region is now greater than 85%, with only Vanuatu and the Solomon Islands achieving 75% and 73% respectively at the completion of the program [55]. However, the proportion of newborns receiving their first dose varies greatly between countries. For example, the majority of newborns in Tonga (92%) received their first dose within 48 h of birth, compared with 42.6% in Kirabati [55]. Low endemicity countries Australia and New Zealand report high rates of infant vaccination coverage (95% and 92% coverage respectively) [53]. However, there are communities within these countries with higher than background hepatitis B prevalence, such as First Nations Peoples [56,57]. The presence of these high-prevalence groups in low endemicity countries emphasizes the importance of maintaining high rates of vaccination coverage.

Similarly, in the Region of the Americas (AMR), reported hepatitis B vaccination coverage is 91% in the United States of America [53]; however, prevalence is higher among key communities including Alaskan First Nations people. In the native Alaskan population, universal infant vaccination was commenced in 1982 and birth dose from 1984 [27]; this program has achieved 93% coverage [58]. Within Latin America, hepatitis B endemicity is low to intermediate and variable between regions, with the highest prevalence in the Amazon Basin region of Brazil. Brazil introduced universal hepatitis B vaccination with birth dose in 1998; however, latest data showed a suboptimal three-dose vaccination coverage of 77% [59,60]. Colombia, similarly, introduced universal vaccination including birth dose in 1992; however, in 2016, estimates of coverage of timely birth dose were only 31% and the timing of doses were not as per the recommended schedule [61].

Within the European Region (EUR), figures from 2017 estimated 91% of infants had received three doses of the hepatitis B vaccine within 24 months of birth, an increase from 81.9% in 2009. As of 2018, hepatitis B vaccination was mandatory in ten European union nations: France, Hungary, Slovakia, Slovenia, Poland, Latvia, Italy, Bulgaria, Croatia and the Czech Republic; and recommended in the remaining countries [62]. In the United Kingdom, latest data shows 93% three-dose vaccine coverage of infants. The universal vaccination schedule was updated to include hepatitis B in 2017 [63]; however, birth dose is only offered to infants born to mothers identified to have chronic hepatitis B through the antenatal screening program, which may not identify all those at risk, presenting an ongoing public health challenge for universal coverage [64].

The Eastern Mediterranean region (EMRO), which comprises 22 countries from North Africa, the Middle East, Eastern Europe and Central Asia, also received support from Gavi for the introduction the three infant doses of vaccination, and by 2008, all but one country had implemented universal vaccination, with the final country on board in 2013. The implementation of universal birth dose was staggered in the region and was implemented in 14/22 countries by 2015. Countries with universal birth dose policies had higher numbers and proportions of health institution births, facilitating timely birth dose delivery [65].

Within the African Region (AFRO), the implementation of universal hepatitis B vaccination in sub-Saharan Africa through the EPI was from as early as 1990 in The Gambia to 2007 in Democratic Republic of Congo and Ethiopia, leading to a decrease in prevalence of hepatitis B among children and adolescents [66]. In 2020, median coverage of vaccination in sub-Saharan Africa was 82.5% [53,67]. With the EPI, most sub-Saharan African countries have implemented universal vaccination; however, only nine countries in the African region had introduced the birth dose universally by 2017. This is due to multiple barriers to delivery of birth dose in low-resource settings, including lack of funding for the monovalent birth dose by the GAVI Alliance, who initially only financially supported pentavalent doses administered later in a child’s first year [68]. In a meta-analysis evaluating the coverage of birth dose vaccination of hepatitis B in Sub-saharan Africa, there was a pooled birth dose vaccination rate of rate of 60.8%; however, this was delivered within four weeks of birth, not within 24 h [69]. Data from individual countries shows significant variation in overall dosing and timeliness. Universal vaccination in the Ivory Coast was implemented in 2000 with 80% coverage of birth dose and only 63% of the third dose; however, only 64% of the birth dose and 45% of the third dose were given within recommended timeframes [70]. In Ghana, there was no access to the birth dose, but three doses were given at 6, 10 and 14 weeks. Complete coverage of three hepatitis B vaccine doses <12 months of birth is greater than 95%; however, delays in timely vaccine delivery were reported up to a median of 4 weeks [71].

## 5. Cost-Effectiveness of Hepatitis B Vaccination for Prevention of HCC

Hepatitis B vaccination has also been shown to be highly cost-effective for reducing HCC incidence and deaths. Early cost-effectiveness modelling of universal hepatitis B vaccination in Israel estimated a US$690,000 health services benefit related to HCC-related costs over 45 years (1990–2035) [72]. In China, the cost-utility model of the universal vaccination program in 2016 determined a saving of US$3779.16 per quality-adjusted life years (QALY) with universal vaccination compared with non-vaccination and prevented 18,322.25 QALY due to hepatitis B-related complications. This resulted in an estimated saving of US$347,962,414.43 over the lifespan of the target cohort [73]. Using Taiwanese data, Markov modelling demonstrated that universal vaccination led to societal cost savings of NT$55,201 per year and healthcare-associated cost savings of NT$23,830 per year by reducing long term complications associated with chronic hepatitis B, including HCC [74].

Cost-effective strategies aimed at increasing birth dose vaccination coverage in low-resource settings include off-label, controlled temperature chain (CTC) storage strategies, with vaccinations able to be stored in temperatures of 40 degrees Celsius for up to 4 weeks, as well as the utilization of prefilled delivery devices. This strategy appears cost-effective, with estimated savings up to USD$1920 (1540–2140) in sub-Saharan Africa and 167 (95% CI 136–198) DALYs per 1000 births avoided [75,76,77]. CTC improved coverage of birth doses in areas requiring outreach services for their delivery, with 88% coverage of timely birth dose in a pilot study conducted in the Laos regions. However, there were issues with sustaining these practices with almost half of the facilities depleting stocks of vaccines three months after the pilot study concluded [78]. Such options could also be utilized in areas impacted by conflict, using lay health workers where trained health care workers are inaccessible [65,79]. CTC cost-effectiveness is sensitive to the costs of outreach service delivery and population prevalence: with low levels of HBsAg prevalence, the cost was $79.71 per DALY averted, whereas in central Asia and eastern Europe, the cost was $0.15 per DALY averted [80].

Prefilled, single-use syringes (Uniject) were initially developed to reduce needle stick injuries due to reusable syringes incompletely sterilized prior to repeated use. Uniject vaccination was administered by village midwives who were trained by the Indonesian Health Department to provide community-based maternal and neonatal care. This option was well received by staff and mothers as it was feasible and acceptable. Uniject devices were also cheaper than standard disposable syringes at the health center (US$6.57 compared with US$7.19 per child immunized) and reduced wastage [81,82].

## 6. Challenges Facing Hepatitis B Vaccination Programs

The World Health Organization’s Immunization Agenda for 2030 aims to reduce the incidence of chronic viral hepatitis B infections by 95% and the number of hepatitis-B-related deaths by 65% by 2030, through improved coverage of primary series and birth dose vaccinations [38]. However, several challenges to achieving these elimination targets must be addressed. The inadequacy of current birth dose vaccination coverage of only 42% [36,53,83] remains a major global public health challenge which is greatly impacted by many factors, including geography [84,85], socioeconomic status, parental education level [86,87], gaps in communication and cultural sensitivities [52,88], health care worker awareness [89], health facility birth rates, antenatal care uptake, lack of health service availability outside of business hours, lack of cold chain infrastructure for vaccine storage [90] and clear institutional guidelines about timing and safety of birth dose vaccination [79,91,92]. Beyond cost-effectiveness, affordability remains a key barrier to birth dose vaccination delivery in many countries and is not covered by the initial Gavi program [77].

Vaccine failure must also be considered, although the rate of vaccine non-response in children is low (<2%), supporting early administration of vaccination at birth and infancy [32,70,93,94]. Administration of a fourth vaccination dose improved efficacy in non-responders to three doses [24,93]. Even within the Taiwanese population, with impressive hepatitis B vaccination coverage rates, approximately 10% of children born to HBeAg-seropositive mothers subsequently became HBsAg carriers, emphasizing the ongoing risk of prenatal intrauterine transmission in mothers with high viral loads and the importance of both active and passive immunization at birth with birth dose and HBIg, the delivery of which has also been challenging [52,93,95,96]. Within African nations with high endemicity of HIV, coinfection has also impacted upon vaccination success [28]. Data for direct impact of hepatitis B vaccination failure is limited; however, the risk for HCC development is similar to that of non-vaccinated people [19,24].

Cultural factors impact on access to health care services in the post-partum, particularly when hepatitis B has not been made a health priority and opportunities for antenatal counselling are missed. A recognized contributing factor to general vaccination uptake is the changing perception of vaccine importance due to increased misinformation and anti-vaccination movements leading to vaccine hesitancy in more recent years [62,90,95]. Here, COVID-19 represents an opportunity for non-siloed approaches to vaccination through coupling catch-up hepatitis B vaccination with COVID-19 vaccination delivery programs and general-population-level education strategies to improve general knowledge and acceptability of vaccination.

## 7. Future Directions

The global Investment Framework for hepatitis B recommends building on national health systems to enable scaling up of hepatitis B elimination activities including preventive vaccination, with a focus on investing in community engagement and education of healthcare workers and the public; stakeholder engagement and ongoing scale-up of vaccination coverage, including working with pre-existing maternal health services to optimize resources for birth dose delivery [68,79,83,84,97]; reducing vaccine hesitancy [96,97]; and utilization of mobile phones to improve communication with community workers regarding home births and need for vaccination administration [68,98,99].

Support from Gavi phase three over 2021–2025 for expanding the reach of birth dose vaccination will be an exciting step toward improved affordability and access to birth dose vaccination worldwide. Utilizing some of the strategies outlined above to improve affordability and delivery effectiveness of birth dose vaccination would markedly improve birth dose vaccine access and coverage and thereby reduce chronic hepatitis B prevalence and related HCC incidence and mortality [42,77].

The adoption of and further development of cost-effective strategies, such as with CTC, prefilled uniject devices and other novel devices as they become available (e.g., Microarray patches) will be essential to increase cost-effectiveness and coverage in low resource settings [76,100].

## 8. Conclusions

Universal hepatitis B vaccination has now been shown to reduce HCC incidence in children and young adults. However, optimizing its global uptake and coverage remains a major challenge. Ongoing efforts spearheaded by the World Health Organization and regional special interest groups are vital to achieve 2030 WHO hepatitis B elimination targets and the hepatitis B Immunization Agenda for 2030, with the substantial impact of these successes on hepatitis-B-related HCC prevention in adults to emerge in the coming years.

## Figures and Tables

**Figure 1 vaccines-10-00793-f001:**
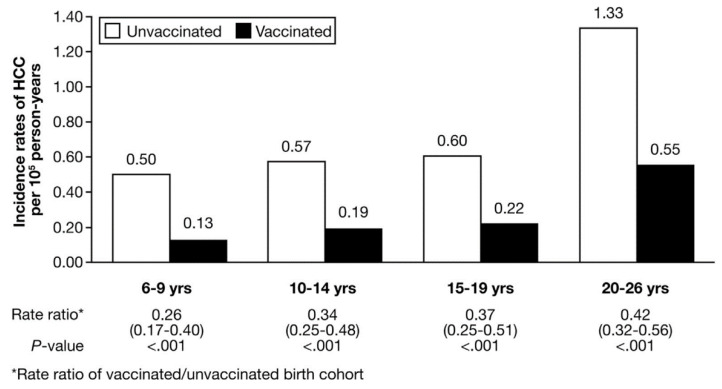
Comparison of HCC incidence rate ratios (95% CI) by age group cohorts born before and after the commencement of the universal hepatitis B vaccination program in Taiwan. Reprinted from Gastroenterology, Vol 151 (3), Chang et al., Long-term Effects of Hepatitis B Immunization of Infants in Preventing Liver Cancer, Pages 472–480.e1, Copyright (2016), with permission from Professor Chang and Elsevier.

**Figure 2 vaccines-10-00793-f002:**
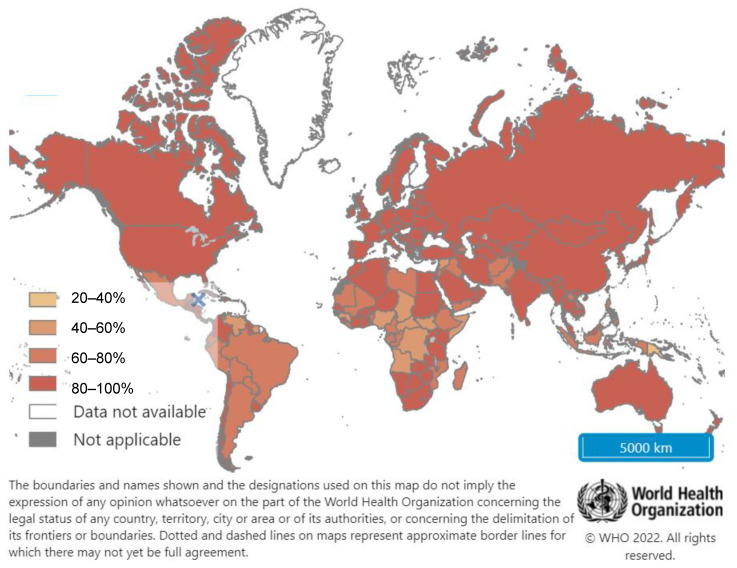
Coverage of three-dose Hepatitis B vaccination among 12-month-olds. Reprinted from Hepatitis B (HepB3) immunization coverage among 1-year-olds (%), Year: Latest data, Copyright WHO (2022). URL: https://www.who.int/data/maternal-newborn-child-adolescent-ageing/indicator-explorer-new/mca/hepatitis-b-(hepb3)-immunization-coverage-among-1-year-olds-(-) [date accessed 15 March 2022]. Reproduced with permission.

**Figure 3 vaccines-10-00793-f003:**
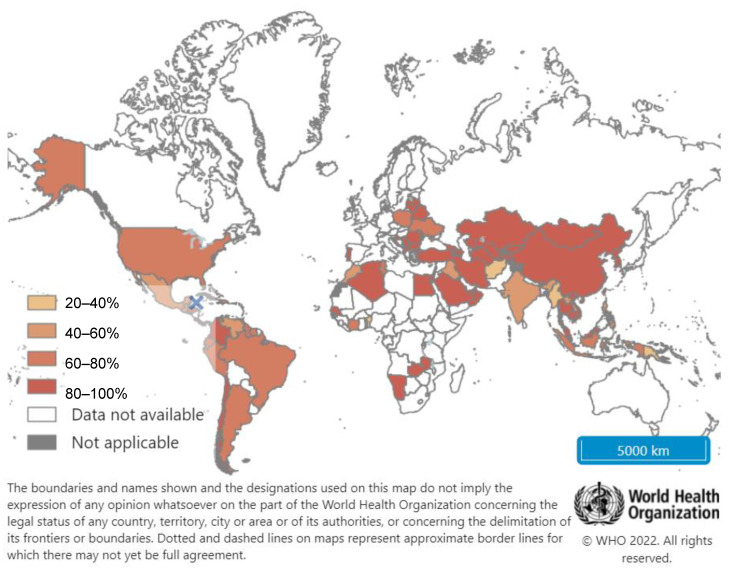
Coverage of first dose Hepatitis B vaccination given within 24 h of birth (%). Reprinted from Hepatitis B first dose vaccine given within 24 h after birth (%), Year: Latest data, Copyright WHO (2022). URL: https://www.who.int/data/maternal-newborn-child-adolescent-ageing/indicator-explorer-new/mca/hepatitis-b-first-dose-vaccine-given-within-24-h-after-birth-(-) [date accessed: 15 March 2022]. Reproduced with permission.

**Table 1 vaccines-10-00793-t001:** Summary of studies assessing the change of incidence, incidence rates and relative risk (RR) related to hepatitis B vaccination programs.

Author	Year	Country/ Region	HCC-Related Outcomes
Chang et al. [16]	1997	Taiwan	Average incidence (/100,000) of HCC declined (trend *p* < 0.01): ○ 1981–1986: 0.7 (range 0.65–0.78) ○ 1986–1990: 0.57 (range 0.48–0.62) ○ 1990–1994: 0.36 (range 0.23–0.48) Incidence rate ratio 1986–1990/1981–1986: 0.63 Age-adjusted RR of HCC after July 1990 compared with before: 0.33 (*p* < 0.001) Age-adjusted risk of death after July 1990 compared with before: 0.51 (*p* < 0.001)
Lee et al. [17]	2003	Taiwan	In age group 0–14 years born in 1996–1999 compared with 1980–1983: ○ Male mortality decreased by 70% ○ Female mortality decreased by 62%
Chang et al. [18]	2009	Taiwan	Comparing HCC incidence from July 1983 to June 2004: Overall RR 0.31 of HCC development in vaccinated group compared with unvaccinated group Incidence rates of HCC in unvaccinated age groups (/100,000py): ○ 6–9 years: 0.49 ○ 10–14 years: 0.56 ○ 15–19 years: 0.60 ○ 20–24 years: 1.07 ○ 25–29 years: 2.28 Incidence rates of HCC in vaccinated age groups (/100,000): ○ 6–9 years: 0.15 ○ 10–14 years: 0.19 ○ 15–19 years: 0.16 Rate ratios of HCC development in vaccinated age groups compared with unvaccinated age groups (all *p* < 0.001) ○ 6–9 years: 0.3 (95% CI 0.18–0.42) ○ 10–14 years: 0.32 (95% CI 0.21–0.49) ○ 15–19 years: 0.30 (95% CI 0.16–0.58) Incomplete vaccination associated with OR 4.32 for HCC development compared with complete vaccination (95% CI 2.34–7.91, *p* < 0.001)
Chang et al. [19]	2016	Taiwan	HCC incidence rates between June 1983 and June 2011 in age group 6–26 years (vaccinated vs. unvaccinated): Overall RR 0.24 Overall incidence rate (/100,000 py): 0.92 vs. 0.23 Incidence rate ratios for vaccinated/unvaccinated age groups (all *p* < 0.0001) ○ 6–9 years: 0.26 (95% CI 0.17–0.40) ○ 10–14 years: 0.34 (95% CI 0.25–0.48) ○ 15–19 years: 0.36 (95% CI 0.25–0.51) ○ 20–26 years: 0.42 (95% CI 0.32–0.56)
Hung et al. [20]	2015	Taiwan	Annual percentage change in age-standardized incidence rates for age groups from 2003 to 2011 (all *p* < 0.05): ○ Children: −16.6% (95% CI −29.7, −1.0,) ○ Adolescents and young adults: −7.9% (95% CI −10.0, −5.7) ○ Middle aged: −2.0% (95% CI −2.8, −1.1) ○ Elderly: 1.3% (95% CI 0.6, 1.9)
Chien et al. [21]	2014	Taiwan	Incidence rate of HCC development (per 100,000 y) according to maternal HBsAg/HBeAg status: ○ HBsAg (−)/HBeAg(−): 0.027 ○ HBsAg (+)/HBeAg(−): 0.162 ○ HBsAg (+)/HBeAg(+): 0.786 Incidence rate of HCC development (per 100,000 y) in mothers HBsAg (−)/HBeAg(−) ○ Vaccination complete: 0.099 ○ Vaccination incomplete: 0.444 Incidence rate of HCC development (per 100,000 y) in mothers HBsAg (+)/HBeAg(+) ○ Ig administered: 0.578 ○ Ig not administered: 1.39 Gender-adjusted HR (95% CI) compared with complete vaccination in HBsAg (−)/HBeAg(−) mothers ○ Vaccination incomplete: 4.4 (1.42–13.65 *p* = 0.0103) ○ Ig administered: 5.51 (2.51–12.080 *p* < 0.0001) ○ Ig not administered: 12.71 (5.6–28.81) *p* < 0.0001)
Liao et al. [22]	2021	Taiwan	HCC incidence RR in individuals aged <30 across the periods compared with period 1 (pre-vaccination): ○ Period 2 (post universal vaccination): 1.15 (95% CI 1.00–1.33) ○ Period 3 (post universal healthcare and screening ultrasounds): 1.4 (95% CI 1.2–1.62) ○ Period 4 (post national viral hepatitis treatment and surveillance program introduced): 0.9 (95% CI 0.76–1.05) HCC incidence RR in period 2 compared with period 1 in individuals aged 10–29 years: 1.49 (95% CI 1.31–1.7, *p* < 0.0001)
Wang et al. [23]	2020	Guangxi, China	HCC-related mortality in 2017–2018: Age-adjusted mortality rate (/100,000) (Χ^2^ = 7.9462, *p* = 0.005) ○ LongAn: 53.3 ○ BinYang: 45.3 Mortality rate (per 100,000) in males ages 20–29 (Χ^2^ = 0.174, *p* = 0.667) ○ LongAn: 2.7 (range 2.5–2.8) ○ BinYang: 4.7 (range 4.6–4.8) ○ RR 1.7 Mortality rate (per 100,000) in males ages ≥ 30 years (Χ^2^ = 1.609, *p* = 0.032) ○ LongAn: 133.5 (range 132.2–134.8) ○ BinYang: 116 (range 115.4–117) ○ RR 0.9 Mortality rate (/100,000) in LongAn in age group 20–29 (Χ^2^ = 5.554, *p* = 0.018) ○ 2004: 7.9 (range 4.4–11.4) ○ 2017–2018: 1.4 (range 0.4–2.4) Mortality rate (per 100,000) in LongAn in age group ≥ 30 (Χ^2^ = 0.0412, *p* = 0.839) ○ 2004: 97 (range 90.6–104.5) ○ 2017–2018: 95.9 (91.3–100.4)
Qu et al. [24]	2014	Qidong, China	HCC incidence in children born in 1985–1990 in 41 rural towns across 6 clusters Primary liver cancer incidence rates (per 100,000) ○ Vaccinated towns: 0.21 ○ Unvaccinated towns: 1.41 ○ HR in vaccinated towns: 0.16 (*p* = 0.0224) Protective efficacy of vaccination 84%
The Gambia Hepatitis Study Group [25]	1987	The Gambia, Africa	For the evaluation of protective effect of vaccination on HCC and CLD of children born during the period of 1986–1990 with stepped wedge design of sequential randomization of EPI teams every three months over four-year period, until all EPI teams administering HBV vaccine with other vaccinations. Long term follow up through the national cancer registry continues. Outcomes not available at time of publication.
Viviani et al. [26]	2008	The Gambia, Africa	65% subjects available for follow up. With expected cumulative incidence based on age-specific HCC incidence rates from 1987–2002, final outcome for detecting significant impact of vaccination on HCC development will be measurable between 2017 and 2020 when subjects are approximately 30 years old. Outcomes not available at time of publication.
McMahon, et al. [27]	2011	Alaska, United States of America	HCC incidence identified by national cancer institute cancer registry and HCC surveillance program set up by Liver Disease and Hepatitis Program between 1969 and 2008: Incidence of HBV-HCC in 1970s was high with 1/3 HCC cases occurring in <30 years Annual incidence of HCC (/100,000) in children aged < 20 years (*p* < 0.001 for overall trend) ○ 1984–1988 (peak): 3 ○ 1999: 0

## Data Availability

Not applicable.

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
