# Peer review of "The Global Impact of Hepatitis B Vaccination on Hepatocellular Carcinoma"

_vaccines, 2022, doi:10.3390/vaccines10050793_

Round 1

Reviewer 1 Report

Flores et al. present a review on hepatitis B vaccination and decreasing incidence of hepatocellular carcinoma (HCC). The authors focus on different key points: the impact of HBV vaccination on HCC incidence in Taiwan, China, Gambia, and Alaska; HBV vaccination situation worldwide, etc. The paper provides interesting, extensive and detailed information on the subject. In the introduction, a mention on the HCC incidence caused by hepatitis C and decrease HCC due to direct-acting antiviral medications (DAA) would help underline the importance of hepatitis B as first cause of HCC, especially in Western countries. The authors should consider to simply Table 1, as most of the information on the Table is repeated in the text. Conversely, a figure comparing reduced incidence/relative risk of HCC among countries and/or vaccinated individuals would be more clarifying. Point 5 (cost-effectiveness) could be merged with point 7 (future directions), where cost-effective strategies are also described. Point 6 title (Challenges to successful vaccination elimination programs) is confusing, what do the authors mean?. Again, the authors may consider to merge point 7 with point 4.2 Global HBV vaccination coverage. As a final recommendation, the authors should consider to shorten the text substantially for the benefit of the reader.

Reviewer 2 Report

The authors have conducted an excellent and comprehensive investigation on the global impact of HB vaccination on hepatocellular carcinoma.  With their thorough survey and study, they clearly demonstrated the successful reduction of HCC with well carried out vaccination program.  The authors are commended for their extensive .

Currently, the vaccination program as yet varies in some geographical regions globally, it is hoped that successful HB vaccination program wold be able to elimiate HB infection and HBV-HCC in the near future.

Reviewer 3 Report

well written review on the impact of HBV vaccination on HCC development
minor changes
1) it would be useful to modify the format of table 1 because it is not
very legible and I suggest to summarize the considerations on HCC related
outcomes 2) please report some data on the rates of non-responders to standard dose
hepatitis B virus vaccine in general population
with also suggestions on
what to do in these cases
